# Post-translational amino acid conversion in photosystem II as a possible origin of photosynthetic oxygen evolution

Yuichiro Shimada [1], Takehiro Suzuki[2], Takumi Matsubara [1], Tomomi Kitajima-Ihara[1], Ryo Nagao [1,3], Naoshi Dohmae [2] & Takumi Noguchi [1]✉

Photosynthetic oxygen evolution is performed at the Mn cluster in photosystem II (PSII). The advent of this reaction on ancient Earth changed its environment by generating an oxygenic atmosphere. However, how oxygen evolution originated during the PSII evolution remains unknown. Here, we characterize the site-directed mutants at the carboxylate ligands to the Mn cluster in cyanobacterial PSII. A His residue replaced for D1-D170 is found to be post-translationally converted to the original Asp to recover oxygen evolution. Gln/Asn residues in the mutants at D1-E189/D1-D342 are also converted to Glu/Asp, suggesting that amino-acid conversion is a common phenomenon at the ligand sites of the Mn cluster. We hypothesize that post-translational generation of carboxylate ligands in ancestral PSII could have led to the formation of a primitive form of the Mn cluster capable of partial water oxidation, which could have played a crucial role in the evolutionary process of photosynthetic oxygen evolution.

[1] Department of Physics, Graduate School of Science, Nagoya University, Furo-cho, Chikusa-ku, Nagoya 464-8602, Japan. [2] Biomolecular Characterization Unit, RIKEN Center for Sustainable Resource Science, 2-1 Hirosawa, Wako, Saitama 351-0198, Japan. [3] Research Institute for Interdisciplinary Science, Okayama University, 3-1-1 Tsushima-naka, Okayama 700-8530, Japan. ✉email: tnoguchi@bio.phys.nagoya-u.ac.jp

In oxygenic photosynthesis, oxygen evolution by water oxidation is carried out in the oxygen-evolving complex (OEC)[1,2], which has an inorganic catalytic core of $Mn_4CaO_5$ (Mn cluster), formed in photosystem II (PSII) through a photoassembly process called photoactivation[3–5] (Fig. 1). The water oxidation not only provides electrons necessary for $CO_2$ fixation, but also plays a significant role in sustenance of the environment and life on Earth as a source of oxygen in the atmosphere. In the OEC, two water molecules are oxidized to one oxygen molecule and four protons through a cycle of five intermediates ($S_0$–$S_4$ states)[1,2], which are advanced by light-induced electron transfer. All extant oxyphototrophs have essentially the identical OEC structure with conserved amino acid residues including ligands to the Mn and Ca ions, i.e., six carboxylate and one histidine ligands from D170, E189, H332, E333, D342, and A344(C-terminus) of the D1 protein and E354 of the CP43 protein (Fig. 1a)[6].

The advent of photosynthetic oxygen evolution on ancient Earth was a key event in the co-evolution of Earth and life. It produced an oxidative atmosphere and promoted the evolution of aerobic life followed by its extensive diversification[7,8]. It is unclear when photosynthetic oxygen evolution originated on Earth; the estimates by geochemical and phylogenetic studies span widely from 3.8 to 2.3 billion years ago[9–16]. Recent evolutionary studies of the PSII proteins by Cardona and coworkers suggested that a functional OEC, nearly identical to that found in extant PSII, existed at a homodimeric photosystem stage before the duplications that led to the evolution of CP43/D1 and CP47/D2[14,15]. They suggested that this duplication could have occurred at a very early stage during the evolution of life in the early Archean. However, how the ligand environment of the Mn cluster was developed and the water oxidation reaction was optimized in the ancestral PSII remain unknown[15,17–19].

We recently characterized a mutant of the cyanobacterium *Synechocystis* sp. PCC 6803, in which one of the ligands to the Mn cluster, D1-D170, was replaced with His (D1-D170H), using liquid chromatography-mass spectrometry (LC-MS) and Fourier transform infrared (FTIR) spectroscopy[20]. We found that the His residue at position 170 was converted to the original Asp residue to form normal OEC and restore oxygen evolution during phototrophic growth. Although it was confirmed that this amino-acid conversion occurred at least after transcription[20], its exact mechanism has yet to be clarified.

Here, we investigate the mechanism of the amino-acid conversion in mutants at the ligands to the Mn cluster using LC-MS and FTIR analyses. We analyze the PSII from D1-D170H cells incorporated with isotope-labeled histidine to clarify whether the conversion occurs at the protein level after translation. Furthermore, we examine the conversion in mutants at other carboxylate ligands, D1-E189 and D1-D342. The obtained results provide insights into the evolutionary process of photosynthetic water oxidation and the origin of oxygen in the ancient atmosphere.

## Results

**Post-translational conversion of His to Asp**. The mechanism of the His→Asp conversion in the D1-D170H mutant was first examined at the RNA level. The sequence of *psbA2* mRNA obtained from mixotrophically grown D1-D170H cells was determined from its cDNA. The codon for the amino acid at position 170 was CAT for His, and no trace of a codon for Asp (GAT/GAC) was detected at this position (Supplementary Fig. 1). This indicates that no modification occurred in the mRNA of the *psbA2* gene after transcription in D1-D170H cells.

Modification at the protein level was next examined using isotope-labeled histidine. D1-D170H cells were incorporated with $^{13}C_6$-labeled histidine and isotope labeling of the Asp residue after the conversion was examined. Flash-induced FTIR difference spectra upon the $S_1 \rightarrow S_2$ transition ($S_2/S_1$ difference) measured with the PSII complexes from D1-D170H cells incorporated with unlabeled ([$^{12}C$-His]D170H) and $^{13}C_6$-labeled ([$^{13}C$-His]D170H) histidine were compared (Fig. 2a). Clear $^{13}C$-induced changes were observed in the prominent bands in 1600–1500 and 1450–1300 $cm^{-1}$, typical regions of the asymmetric and symmetric $COO^-$ stretching vibrations, respectively, of the carboxylate groups around the Mn cluster[21]. The $^{12}C$-minus-$^{13}C$ double difference spectrum more clearly revealed the changes showing bands at 1586/1566/1551/1529 $cm^{-1}$ and 1399/1376/1362/1320 $cm^{-1}$, suggesting the $^{13}C$-induced downshifts of differential signals in the $S_2/S_1$ difference spectrum. The spectral feature in the symmetric $COO^-$ region was well reproduced by the quantum mechanics/molecular mechanics (QM/MM) calculations (Fig. 2b), in which the carbon atoms of D1-D170 were selectively labeled with $^{13}C$. The calculated normal modes (Fig. 2b, sticks) revealed that the experimental bands at 1399($S_1$)/1362($S_2$) $cm^{-1}$ and 1376($S_1$)/1320($S_2$) $cm^{-1}$ are attributed to the vibrations of unlabeled and $^{13}C$-labeled D170, respectively.

The $^{13}C$ labeling of D1-D170 converted from His was further examined by LC-MS analysis (Fig. 2c, d). In both [$^{12}C$-His]D170H and [$^{13}C$-His]D170H cells, the H170 residue in 58–59% and 3–4% of the D1 proteins was converted to Asp and Asn, respectively (Fig. 2d). In the D1 protein from [$^{13}C$-His]D170H cells, the MS spectral peaks of $^{13}C_4$-labeled D170 (Fig. 2c) and N170 (Supplementary Fig. 2b) in addition to the peak of $^{13}C_6$-labeled H170 (Supplementary Fig. 2a) were detected. The MS chromatogram bands (Fig. 2c and Supplementary Fig. 2) showed that 66% of the His, Asp, and Asn residues at D1-170, in total, was labeled with $^{13}C$ (Fig. 2d), which is similar to ~61% $^{13}C$ labeling at D1-H332 as a histidine ligand to the Mn cluster and at D1-H190 interacting with $Y_Z$, indicating that 60–70% of His residues were labeled with $^{13}C$ in [$^{13}C$-His]D170H cells. It is

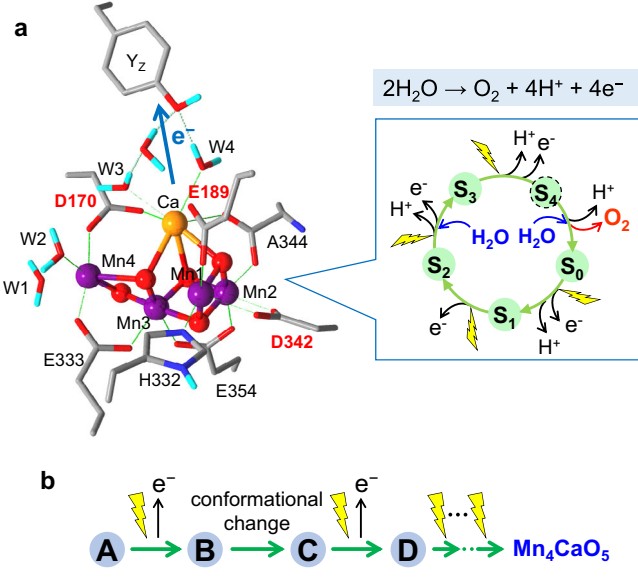

**Fig. 1 Structure and reaction of the oxygen-evolving Mn cluster and its photoassembly process. a** Structure of the Mn cluster and its ligands, which was obtained by QM/MM calculation based on the X-ray crystallographic structure[6]. Mn, purple; Ca, orange; O, red; N, blue; and H, cyan. The amino acid residues are on the D1 protein except for E354 on the CP43 protein. D1-D170, E189, and D342 are the sites of mutations in this study. The inset shows the S-state cycle of water oxidation. **b** Early process of the photoassembly of the Mn cluster.

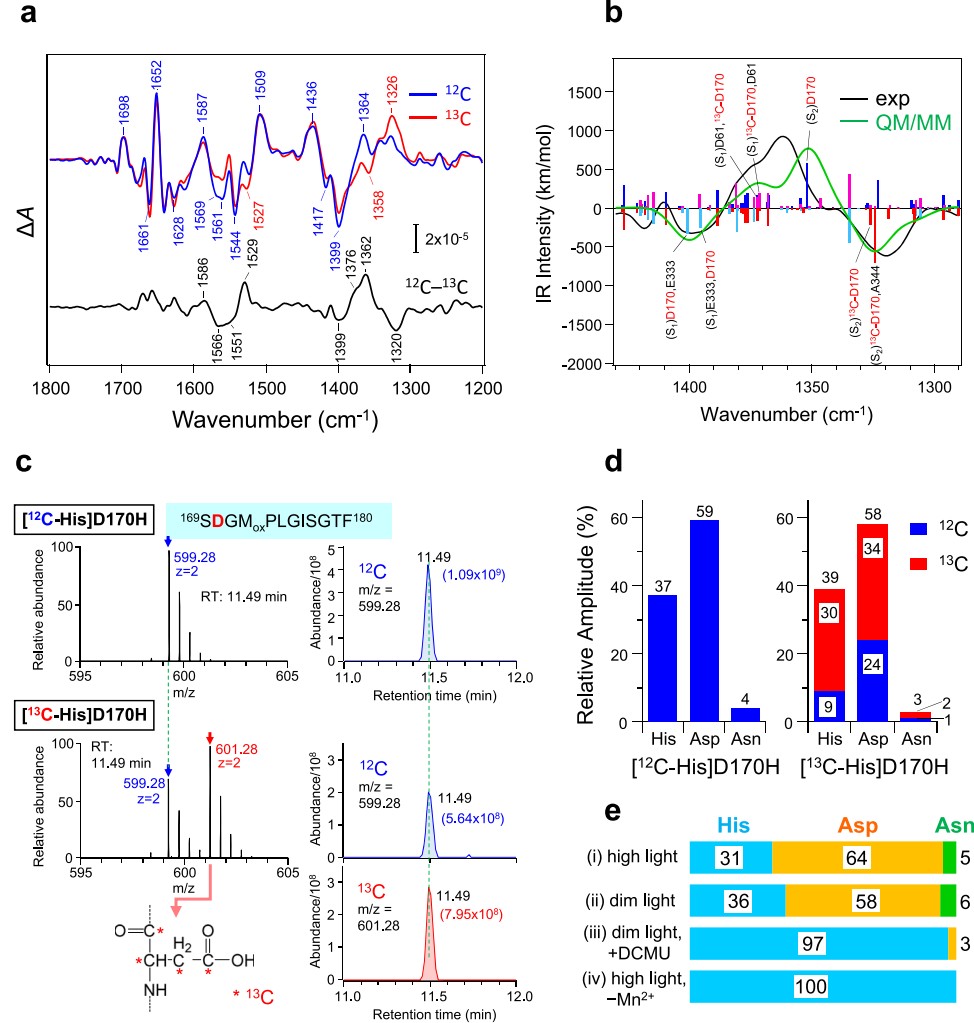

**Fig. 2 FTIR and LC-MS analyses of the D1-D170H mutant. a** Flash-induced $S_2/S_1$ FTIR difference spectra of the PSII complexes from D1-D170H cells incorporated with unlabeled ([$^{12}$C-His]D170H: blue line) and $^{13}C_6$-labeled histidine ([$^{13}$C-His]D170H: red line), together with a $^{12}$C-minus-$^{13}$C double difference spectrum (black line). **b** The $^{12}$C-minus-$^{13}$C infrared difference spectrum simulated by QM/MM calculations (green line) in comparison with the experimental spectrum (black line). The normal modes of the unlabeled ($S_1$: cyan sticks; $S_2$: blue sticks) and $^{13}$C-D170-labeled ($S_1$: magenta sticks; $S_2$: red sticks) OEC are superimposed. **c** MS spectra (left) and MS chromatograms (right) of the polypeptide fragments of the D1 proteins (between S169 and F180 with oxidized M172) in [$^{12}$C-His]D170H and [$^{13}$C-His]D170H cells. The data of the D1 fragment that has Asp at position 170 are shown. A peak of the target polypeptide fragment in the MS spectrum is indicated by a blue ($^{12}$C-D170) or red ($^{13}$C-D170) arrow. The area intensity of the chromatogram peak is given in parentheses. **d** LC-MS estimation of the relative amounts of the D1 fragments having His, Asp, and Asn residues at position 170 with $^{12}$C (blue) and $^{13}$C (red) atoms in [$^{12}$C-His]D170H (left) and [$^{13}$C-His]D170H (right) cells. **e** LC-MS estimation of the relative amounts of the D1 fragments with His, Asp, and Asn residues at position 170 in D1-D170H cells grown with glucose under (i) high light, (ii) dim light, (iii) dim light in the presence of DCMU, and (iv) high light in the absence of $Mn^{2+}$. Data of (i) and (iii) are taken from Kitajima-Ihara et al.[20]. Source data are provided as a Source Data file.

noted that no appreciable conversion of the D1-H332 ligand to Asp was detected.

Thus, from the FTIR, QM/MM, and LS-MS results together with the RNA analysis, it is definitely concluded that D1-H170 in the D1-D170H mutant was post-translationally converted to the original Asp residue (and to Asn to a minor extent) at the protein level.

**Involvement of light-induced oxidation of Mn.** We previously showed that phototrophic growth is necessary for this His→Asp conversion[20]. Although under heterotrophic condition with herbicide DCMU and dim light, the His→Asp conversion hardly occurred (~3%; Fig. 2e, iii)[20], the same dim-light condition in the absence of DCMU converted ~58% of H170 to Asp (Fig. 2e, ii and Supplementary Fig. 3), indicating the necessity of electron transfer in PSII for the conversion.

The involvement of a Mn ion(s) in the amino-acid conversion was (were) further examined. It was previously reported that *Synechocystis* sp. PCC 6803 cells can grow in $Mn^{2+}$-depleted medium keeping the PSII level[22]. We grew D1-D170H mutant cells under mixotrophic condition in $Mn^{2+}$-depleted medium. The PSII complexes prepared from the thus obtained cells showed no $O_2$ evolution and LC-MS analysis detected only His at D1-170 without any trace of Asp (Fig. 2e, iv and Supplementary Fig. 4). These observations thus indicate that a $Mn^{2+}$ ion(s) photo-oxidized in the OEC site is (are) directly involved in the mechanism of the His→Asp conversion.

**Conversion at other ligand sites and amino acid dependence.** We further examined the amino acid conversion at other ligand sites of the Mn cluster and dependence on the amino acid species. D1-E189 and D1-D342 were replaced with corresponding amide

**a**

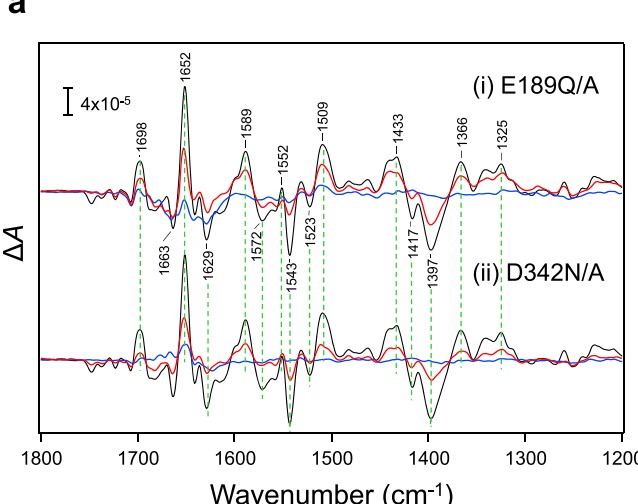

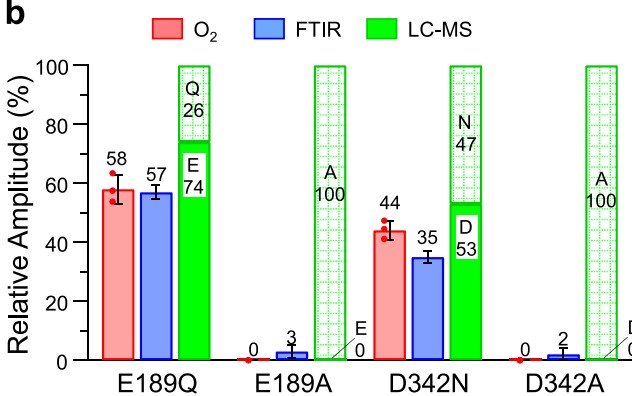

**b**

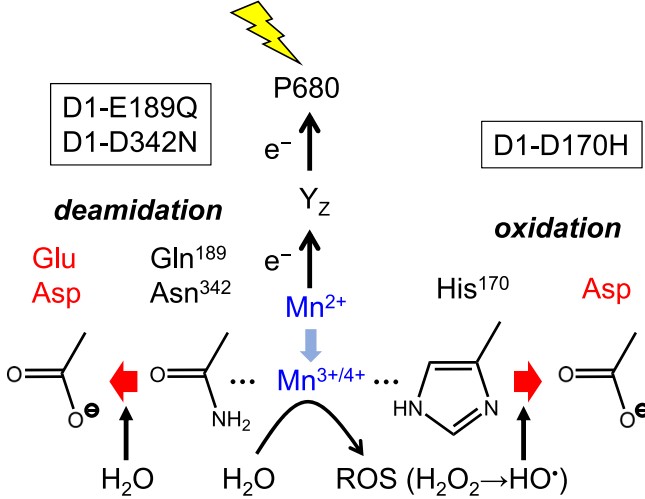

**Fig. 4 Possible mechanisms of post-translational amino acid conversion to form carboxylate ligands.** His in the D1-D170H mutant is oxidized to Asp by reactive oxygen species (ROS) such as a hydroxyl radical (HO•) produced from $H_2O_2$, which is formed by partial oxidation of water at the $Mn^{3+}/Mn^{4+}$ site. Gln and Asn in the D1-E189Q and D1-D342N mutants are converted to Glu and Asp, respectively, by deamidation.

original Glu/Asp residues, whereas such conversion was detected neither in the E189A nor D342A mutants (Fig. 3b). The slightly lower $O_2$ evolution and FTIR-estimated $S_1 \rightarrow S_2$ activities than the extents of conversion estimated by the LC-MS analysis in E189Q and D342N cells (Fig. 3b) may be ascribed to the presence of some inactive centers in the isolated PSII core complexes with converted residues.

These results indicate that the amino-acid conversion to the original carboxylate residues is a common phenomenon at the ligand sites of the Mn cluster, although the capability of conversion depends on the amino acid species. His/Asn and Gln side chains have enough numbers of carbon atoms for conversion into Asp and Glu, respectively, whereas Ala cannot be changed to an Asp/Glu residue without insertion of a carbon atom(s).

## Discussion

In this study, we found that some amino acid residues at the ligand sites of the Mn cluster can be post-translationally converted to the original carboxylate residues. An oxidized Mn ion(s) produced by light-induced electron transfer is (are) most likely involved in the mechanism of this amino-acid conversion. In the photoactivation process, the first $Mn^{2+}$ ion, probably bound to D1-D170 and D1-E189[5,25], is oxidized to $Mn^{3+}$, and then after the conformational change of the D1 C-terminal region, involving D1-E333 and D1-D342[5], another $Mn^{2+}$ is oxidized to $Mn^{3+}$ to form a relatively stable binuclear complex[3,4]. Further illumination oxidizes Mn ions to $Mn^{3+/4+}$ to finally construct the Mn cluster with a $Mn_4CaO_5$ form (Fig. 1b).

Some cases of post-translational amino-acid conversion have been reported in vivo and in vitro[26–29]. One typical case is metal-catalyzed oxidation of a His residue by a hydroxyl radical induced by $H_2O_2$ with an aid of a metal ion, typically $Fe^{2+}$ or $Cu^+$, through Fenton-like reactions[26,27]. His is first converted to 2-oxo-histidine and then to Asp or Asn. $Mn^{2+}$ is also effective in producing a hydroxyl radical from $H_2O_2$[30], and some Mn oxide complexes supplemented with $H_2O_2$ have been reported to oxidize various organic compounds[31,32]. Indeed, $H_2O_2$ has been detected as a byproduct in water oxidation reactions of some modified OEC[33]. It is thus speculated that reactive oxygen species

**Fig. 3 FTIR and LC-MS analyses of the D1-E189Q/A and D1-D342N/A mutants. a** $S_2/S_1$ FTIR difference spectra of the PSII complexes from (i) D1-E189Q (red line) and E189A (blue line) and (ii) D1-D342N (red line) and D342A (blue line) cells in comparison with the spectrum of WT* PSII (black lines). The amplitudes of the spectra were normalized based on the protein amounts estimated from the amide II band. **b** Comparison of the $O_2$ evolution activity (red bars) and the FTIR amplitude ($\Delta A$ between 1433 and 1397 cm⁻¹; blue bars) relative to those of WT* and the relative contents of amino acid residues at the mutation sites estimated by LC-MS analysis (green bars) in the PSII complexes from D1-E189Q, E189A, D342N, and D342A cells. The $O_2$ evolution activities were presented as mean values ± standard deviation ($n = 3$), and the error of the FTIR amplitude was estimated from the root-mean-square noise level in the 1400–1390 cm⁻¹ region of the dark-minus-dark spectrum. Source data are provided as a Source Data file.

residues (E189Q and D342N) and a smaller Ala residue (E189A and D342A). The PSII complexes from E189Q and D342N cells grown mixotrophically showed $O_2$ evolution activities of 58 ± 3 and 44 ± 5% of the WT* PSII, whereas those from E189A and D342A cells showed no $O_2$ activity (Fig. 3b). The $S_2/S_1$ FTIR difference spectra of the E189Q and D342N PSII (Fig. 3a) showed features very similar to those of the WT* PSII, which is consistent with the previous results[23,24], but with amplitudes of 57 and 35%, respectively, on the protein basis (Fig. 3b). In contrast, the FTIR spectra of the E189A and D342A PSII showed almost no intensities (Fig. 3a, b).

The LC-MS analysis of the D1 protein from these mutants (Fig. 3b, Supplementary Figs. 5 and 6) further showed that in D1-E189Q and D1-D342N cells, 74 and 53% of the D1 proteins, respectively, underwent conversion from Q189/N342 to the

(ROS) such as $H_2O_2$ and a hydroxyl radical produced from water during photoactivation oxidizes a His residue at the ligand site to Asp or Asn (Fig. 4). Another case is deamidation of Asn and Gln residues to change into Asp and Glu, respectively[28,29]. Deamidation is carried out by hydrolysis of amide groups to carboxylate groups, and in proteins the rate of deamidation strongly depends on the primary sequence and the 3D structure around the reacting residue[28]. A relationship was also suggested between metal binding and deamidation in amyloidogenic proteins[29]. It is thus possible that Mn binding at amide residues promoted deamidation to generate Glu/Asp ligands (Fig. 4).

The amino acid residues that are converted to carboxylate residues may not be limited to His and amide residues. It has been reported that mutations of D1-D170 to Tyr, Trp, Asn, His, Met, Glu, Arg, Val, Leu, and Ilu, all of which have long side chains at least with a γ-carbon, retained partial $O_2$ evolution activity[34–36], suggesting the presence of the intact Mn cluster in some PSII complexes in the mutants (note that Glu replaced for D1-D170 may form a partially functional Mn cluster[37]). In contrast, mutations of D1-D170 to shorter amino acid residues such as Ala and Ser did not show $O_2$ evolution[34,35]. In addition, mutations of D1-E189 to Gln, Arg, Lys, Leu, and Ile, which have an aliphatic δ-carbon, largely restored the $O_2$ evolution, whereas mutations to residues having shorter aliphatic chains showed no or only a trivial amount of $O_2$ evolution[36,38]. Furthermore, mutations of D1-D342 to Asn, His, and Glu having a γ-carbon showed some $O_2$ evolution[35,39], whereas the Ala mutant did not support $O_2$ evolution[39] in agreement with our result (Fig. 3). These observations can be explained by post-translational conversions from residues with side chains long enough to form Asp/Glu and the absence of conversion from those with shorter side chains. In particular, the presence of $O_2$ activity upon mutations of D170 or E189 to aliphatic residues such as Val, Lue, and Ile[36,38], which cannot ligate the Mn cluster, is understandable if these side chains are converted to the Asp/Glu side chains. These amino-acid conversions except for deamidation of Asn/Gln may also be promoted by ROS formed from water through $Mn^{3+/4+}$ ions, although the exact conversion mechanism for each amino acid residue requires further investigations.

In the D170H, E189Q, and D342N mutants, relatively large amounts (>50%) of the D1 proteins with converted original amino acid ligands were accumulated in mixotrophically grown cells (Figs. 2 and 3). In the steady state of the turnover of the D1 protein in the cell culture of the mutants, where the total amount of the D1 protein is constant and the relative amounts of the converted functional and initial unconverted D1 proteins ([cD1] and [iD1], respectively) are unchanged, their ratio ([cD1]/[iD1]) is determined by the rate constant of conversion ($k_c$) relative to that of degradation of the converted D1 protein ($k_2$) ([cD1]/[iD1] = $k_c/k_2$; Supplementary Fig. 7). The turnover of the D1 protein in the *Synechocystis* sp. PCC 6803 mutant with *psbA2* as a single gene for the D1 protein (corresponding to our WT*) was previously shown to occur in several hours under the light of 100 μmol photons $m^{-2}s^{-1}$ [40]. Thus, the presence of comparable amounts of the converted and unconverted D1 proteins indicates that amino-acid conversion in the D170H, E189Q, and D342N mutants through ROS-mediated oxidation or deamidation has time constants of hours under the light of ~50 μmol photons $m^{-2}s^{-1}$ used in our experiments. In the oxidation by ROS, a variety of oxidized products, such as 2-oxo-histidine in the His→Asp conversion, would be formed as impaired D1 proteins. However, because non-functional D1 proteins degrade much faster than the functional D1 protein (i.e., large $k_2$)[40,41], they are hardly accumulated in cells. For the same reason, the D1 protein modified by the His→Asp conversion at D1-H332, an original His ligand to the Mn cluster, will not be accumulated in cells, because of the formation of a non-functional D1 protein[39].

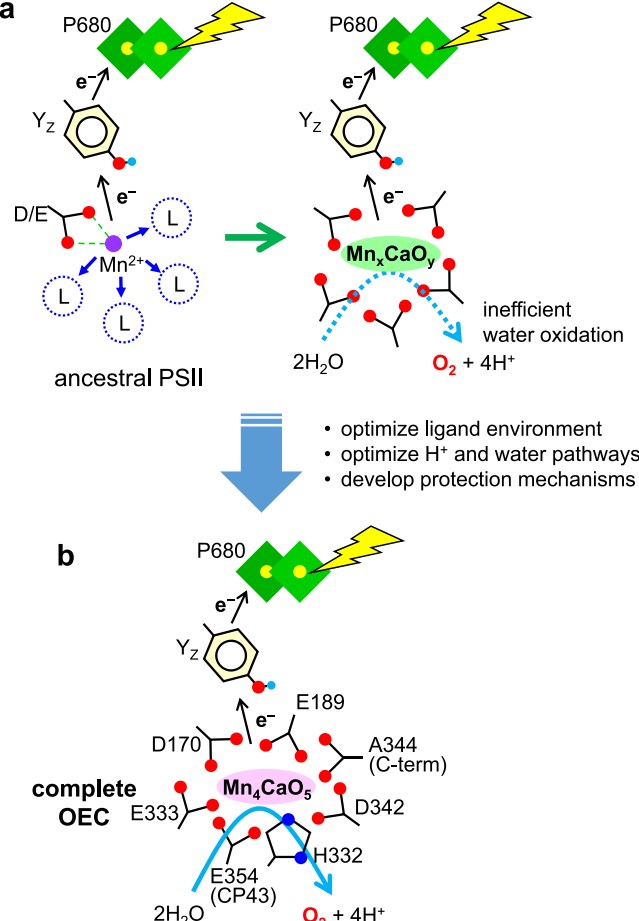

**Fig. 5 Hypothetical mechanism of generation of a primitive OEC in ancestral PSII. a** The ancestral PSII had a D/E residue(s) for binding of an initial $Mn^{2+}$ ion, which can be oxidized by light-induced electron transfer. Carboxylate ligands were generated by post-translational amino-acid conversion and some sort of Mn-oxide complex was formed in the OEC site to perform partial water oxidation. During this inefficient $O_2$ evolution, the ligand environment and proton and water pathways were optimized, and protection mechanisms against ROS were developed in PSII. "L" indicates non-carboxylic amino acid residues at the ligand positions. **b** Completed OEC capable of efficient water oxidation.

Post-translational amino-acid conversions so far reported inactivate proteins in relevance to aging and diseases[27–29]. The amino-acid conversion found in this study, which generates carboxylate ligands in OEC, is a very unique case that restores an enzymatic function, photosynthetic oxygen evolution, essential for life and its evolution.

One of the biggest questions in the evolution of photosynthesis is how the OEC originated in the ancestral PSII[15,17–19]. Our data suggest that water oxidation could have originated before a fully complete ligand sphere had time to evolve in the early photosystem. We speculate that in the ancestral PSII, which has at least a single Mn binding site by an Asp/Glu residue(s), a $Mn^{3+/4+}$ ion formed by light-induced electron transfer in the reaction center promoted post-translational conversion of nearby amino-acid residues to generate Asp/Glu ligands (Fig. 5). Such an initial metal binding site on the electron-donor side could have existed even in reaction centers at a very early stage of photosynthesis evolution. Indeed, evidence for such an ancient site may be found in extant homodimeric type I reaction centers. In the *Heliobacterium modesticaldum* type I reaction center, an exposed Ca-binding site was recently discovered with structural similarities to the OEC in

PSII, including two carboxylate ligands to the Ca atom, an Asp and the C-terminal carboxylic group[42,43]. The presence of several carboxylate ligands with negative charges is significant for the formation of a functional Mn cluster not only by fixing the Mn/Ca ions but also by tuning its redox potential. Recently, Chernev et al.[19] showed light-induced formation of birnessite-type Mn(III,IV)-oxide nanoparticles in apo-PSII, and proposed that such a Mn-oxide particle was down-sized to develop the Mn cluster. Thus different types of high-valent Mn-oxide complexes in various sizes could be formed in the OEC site by light-induced electron transfer during the evolution of PSII. It has also been reported that many high-valent Mn-oxide complexes function as catalysts of water oxidation[44]. It is thus presumed that some sort of photo-produced Mn-oxide complex was stabilized and functionalized by post-translationally generated carboxylate ligands in the ancestral PSII, and carried out partial oxygen evolution.

This inefficient water oxidation, however, would have been crucial for further development of PSII; in addition to hetero-dimerization and optimization of the ligand environment, the proton-exit and water-delivery pathways would have been formed around the Mn cluster to optimize the water oxidation reaction, while various protection mechanisms and repair systems would have been developed[14,45]. It would also have stimulated the evolution of aerobic respiration and oxygen-related enzymes[46–48]. Thus, the post-translational amino-acid conversion, beyond the central dogma, to generate the carboxylate ligands to the Mn cluster could have played a significant role in the evolution of photosynthetic water oxidation and that of aerobic life.

## Methods

**Construction of site-directed mutants**. Site-directed mutagenesis of the *psbA2* gene encoding the D1 subunit was performed in *Synechocystis* sp. PCC 6803[49,50]. Plasmid pRN123, which involved the coding region of *psbA2*, was used as a parental vector for site-directed mutagenesis. The host *Synechocystis* strain, which lacks all of the three *psbA* genes (Δ*psbA1*/Δ*psbA2*/Δ*psbA3*) and contains a hexahistidine tag attached to the C-terminus of the CP47 protein, was transformed with pRN123 to provide a control strain with the wild-type D1 protein (WT*). Mutation of D1-D170 to His[20], D1-E189 to Gln/Ala, and D1-D342 to Asn/Ala were introduced into pRN123 by replacing the GAT (D170), GAG(E189), and GAC(D342) codons with CAT (His), CAG(Gln)/GCG(Ala), and AAC(Asn)/GCC(Ala) codons, respectively, and the obtained plasmids were introduced into the host Δ*psbA1*/Δ*psbA2*/Δ*psbA3* strain. These strains were maintained on BG-11[51] agar plates containing antibiotics (kanamycin, chloramphenicol, erythromycin, and spectinomycin, 5 μg mL$^{-1}$ for each), in the presence of 5 mM glucose and 10 μM 3-(3,4)-dichlorophenyl-1,1-dimethylurea (DCMU) under a continuous low-light condition.

**Cell culture and sample preparations**. WT* and mutant cells were grown mixotrophically in 40 mL of BG-11 medium, which was supplemented with 5 mM glucose and the above mentioned antibiotics (5 μg mL$^{-1}$ for each) and was bubbled with air containing 3% (v/v) $CO_2$, at 30 °C under continuous illumination (~20 μmol photons m$^{-2}$s$^{-1}$). For preparation of PSII core complexes, cells were grown mixotrophically in an 8 L culture bottle without antibiotics under continuous illumination (~50 μmol photons m$^{-2}$s$^{-1}$). For a low-light condition, dim light with an intensity of ~5 μmol photons m$^{-2}$s$^{-1}$ was used. Cells cultured in three bottles (total volume of 24 L) were used for preparation of PSII core complexes from each strain. An aliquot of the cell culture at this stage was saved and used for DNA sequencing[20], which showed the proper genotypes of individual mutants and confirmed no trace of the wild-type *psbA2* gene.

For cell growth in a Mn$^{2+}$-depleted medium, the BG-11 medium in the absence of MnCl$_2$·4H$_2$O was prepared with high-grade reagents. In addition to removing MnCl$_2$, we replaced ammonium ferric citrate, which was found to contain high Mn$^{2+}$ contamination, in the original BG-11 medium with FeSO$_4$·7H$_2$O (special grade) and citric acid.

PSII core complexes were purified using Ni$^{2+}$ affinity column chromatography[49]. Thylakoid membranes suspended in a buffer (pH 6.0) containing 50 mM Mes-NaOH, 5 mM CaCl$_2$, 10 mM MgCl$_2$, and 25% (w/v) glycerol (buffer A) were solubilized with 0.8% (w/v) *n*-dodecyl β-D-maltoside (DM) at a Chl concentration of 1.0 mg mL$^{-1}$ by stirring for 20 min on ice. After centrifugation at 43,000 *g* for 10 min, the resultant supernatant was applied to a Ni$^{2+}$ affinity column, which was then washed with buffer A containing 0.04% DM and 1 mM L-histidine. PSII complexes were eluted with buffer A containing 0.04%

DM and 50 mM L-histidine and then concentrated by ultrafiltration (AmiconUltra-15, NMWL 100,000).

The O$_2$ evolution activities of PSII complexes were measured using a Clark-type oxygen electrode at 30 °C under saturating light in a buffer containing 50 mM Mes-NaOH (pH 6.0), 1 M sucrose, 10 mM NaCl, 5 mM CaCl$_2$, 0.04% *n*-dodecyl β-D-maltoside (DM) in the presence of 4 mM potassium ferricyanide and 0.1 mM 2,6-dichloro-1,4-benzoquinone (DCBQ) as exogenous electron acceptors. Values by three measurements on distinct samples were averaged for each mutant.

**Incorporation of $^{13}C_6$-labeled histidine into D1-D170H cells**. For incorporation of $^{13}C_6$-labeled histidine into D1-D170H cells, a histidine-tolerant strain was first isolated[52] using the D1-D170H mutant constructed previously[20]. D1-D170H cells were repeatedly cultured on a BG-11 agar plate with the above antibiotics supplemented with L-histidine, the content of which was increased from 30 μM to 240 μM in a stepwise way. This strain was maintained on a BG-11 agar plate in the presence of 120 μM L-histidine. Cells of this strain were grown in liquid BG-11 medium as described above except for the presence of 120 μM unlabeled ($^{12}C$) or $^{13}C_6$-labeled (Cambridge Isotope Laboratories, Inc., 97–99 at. % $^{13}C$) L-histidine. Cells cultured in two bottles (total volume of 16 L) were used for preparation of PSII core complexes.

**Sequencing of the *psbA2* mRNA from D1-D170H cells**. Total RNA was isolated from mixotrophically grown D1-D170H cells using the TRIzol Max Bacterial RNA Isolation Kit (Thermo Fisher Scientific). Genomic DNA was digested by DNase using the ReliaPrep RNA Cell Miniprep System (Promega). The cDNA of the *psbA2* mRNA was obtained by reverse transcription-PCR using ReverTra Plus (Toyobo) with a primer 5′-GTCAAAGCCACAGGAGCTTGCTCCCC-3′. From the obtained cDNA, a DNA fragment (1.0 kbp) in the internal region of *psbA2* was amplified by PCR using a primer set of 5′-AGCGCGAAAGCGCTTCCTTGTGGG-3′ and 5′-GTCAAAGCCACAGGAGCTTGCTCCCC-3′. The DNA sequence including the amino acid residue at position 170 was determined with a primer 5′-CGGCATTTTCTGCTACATGGG-3′ using an ABI 3100 DNA sequencer.

**Measurement of flash-induced FTIR difference spectra**. For FTIR measurements, PSII suspension (~2 mg of Chl mL$^{-1}$, 10 μL) in a buffer (10 mM Mes-NaOH, 5 mM NaCl, 5 mM CaCl$_2$, 40 mM sucrose, and 0.06% DM, pH 6.0) was added with 1 μL of 100 mM potassium ferricyanide and was dried on a CaF$_2$ plate under N$_2$ gas. For the PSII samples of D1-E189Q/A and D1-D342N/A as well as their control WT* sample, 1 μL of 20 mM NaHCO$_3$ was further added to the suspension to prevent pre-oxidation of the non-heme iron, whose signals often contaminate the spectra. The dried film was sealed in a cell formed by another CaF$_2$ plate with a greased Teflon spacer (0.5 mm in thickness), enclosing 2 μL of 40% (v/v) glycerol solution without touching the sample to moderately hydrate the film[53]. The sample temperature was kept at 10 °C by circulating cold water through a copper holder.

FTIR difference spectra were measured using a Bruker Vertex 80 spectrophotometer (with the OPUS 7.8 software) with an MCT detector at 4 cm$^{-1}$ resolution[20]. A Ge filter (Andover, 4.50ILP-25) was placed in the infrared path to cut light at >2200 cm$^{-1}$. FTIR spectra with 20 s scans were recorded twice before and once after single-flash illumination from a Nd:YAG laser (Quanta-Ray INDI-40-10; 532 nm, ~7 ns fwhm, ~7 mJpulse$^{-1}$ cm$^{-2}$). The measurement was repeated 100 times for D1-D170H and 160 times for D1-E189Q/A, D1-D342N/A, and WT* with a dark interval of 2 min, and spectra were averaged. A difference spectrum of after-minus-before illumination provided the changes upon the S$_1$ → S$_2$ transition, while that of the two spectra before illumination represented the noise level.

**Mass spectrometry analysis**. Amino acid sequences of the D1 proteins were analyzed using mass spectrometry[20]. PSII core complexes solubilized with 3% lithium lauryl sulfate in the presence of 75 mM dithiothreitol were subjected to SDS − PAGE. The isolated D1 protein was applied to nano-liquid chromatography-tandem mass spectrometry (nLC-MS/MS) after digestion with chymotrypsin. The digestion mixture was separated using a nano-electrospray column (NTCC analytical column, C18, φ75 μm × 100 mm, 3 μm; Nikkyo Technos, Tokyo, Japan) on a nanoflow LC (Easy nLC 1000; Thermo Fisher Scientific, Inc., Waltham, MA, USA) with a linear gradient of 0–60% of 0.1% formic acid/acetonitrile in 0.1% formic acid solution at a flow rate of 300 nL/min over 30 min. The LC was coupled on-line to a Q-Exactive mass spectrometer (Thermo Fisher Scientific, Inc.) equipped with a nanospray ion source. A data-dependent TOP 10 method in positive ion mode was used to operate the mass spectrometer. Obtained MS/MS spectra were searched using the MASCOT 2.7 program (Matrix Science, London, UK) against an in-house database containing the D1 protein and its mutants. FreeStyle 1.3 SP2 and Xcalibur 4.1.50, Qual Browser (Thermo Fisher Scientific, Inc.) were used to draw the MS spectra and MS chromatograms.

**Quantum mechanics/molecular mechanics calculations**. The infrared spectra of the carboxylate groups in the OEC were simulated using quantum mechanics/molecular mechanics (QM/MM) calculations[21,54]. The coordinates of heavy atoms in the Mn cluster, surrounding amino acid residues, water molecules, and two Cl$^-$

ions within 20 Å from the Mn cluster were extracted from the X-ray structure (PDB ID: 4UB6)[6] of PSII in the $S_1$ state. Hydrogen atoms were optimized using the AMBER force field[55] fixing all heavy atoms. QM/MM calculations were carried out by the ONIOM method[56] with the electronic embedding scheme in the Gaussian 16 program package[57]. The QM region (Supplementary Fig. 8) consists of the Mn cluster, Cl-1, seven amino acid ligands (D1-D170, D1-E189, D1-H332, D1-E333, D1-D342, D1-A344, CP43-E354) and nearby amino acid residues ($Y_Z$, D1-H190, D1-D61, D1-H337, CP43-R357, D1-N181, D2-K317), and 17 surrounding water molecules including four water ligands (W1–W4), while other atoms were assigned to the MM region. D1-H337 was assumed to have a protonated cation form[58] and W2 was fully protonated $H_2O$[54]. Geometry optimization and normal mode analysis of the QM region were performed using an unrestricted DFT method with the B3LYP functional and the basis sets of LANL2DZ and 6-31 G(d) for metal atoms and other atoms, respectively. In geometry optimization, the coordinates of the QM region were fully relaxed, while those of the MM region were fixed. The oxidation states of the Mn ions were assumed to be $III_2IV_2$ and $III_1IV_3$ in high spin states (15et and 14et) in the $S_1$ and $S_2$ states, respectively. The $S_2$ state was assumed to have an open cubane conformation with oxidized Mn4(IV). In calculation of OEC with [13]C-labeled D1-D170, two carbon atoms of the D170 side chains in the QM region were substituted with [13]C. To generate infrared spectra in the symmetric $COO^-$ stretching region, a Gaussian band with a 16 cm$^{-1}$ width (FWHM) was assumed for each normal mode and all bands in this region were co-added[21,54]. Calculated vibrational frequencies were scaled with a scaling factor of 0.9628 to match the major negative peak of the simulated $S_2/S_1$ spectrum of unlabeled OEC to the experimental peak at 1399 cm$^{-1}$.

**Reporting summary**. Further information on research design is available in the Nature Research Reporting Summary linked to this article.

## Data availability
The data supporting the findings in this study are available within the manuscript and the Supplementary Information file. The sequence of cDNA of the *psbA2* mRNA of D1-D170H was deposited in the DNA Data Bank of Japan under accession number LC717798. The coordinates of the PSII complex used in QM/MM calculations were obtained from PDB ID: 4UB6. Source data are provided with this paper.

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

## Acknowledgements

We thank Drs. Johannes Messinger, Tanai Cardona, Peter J. Nixon, and Nicholas Cox for valuable discussions. The computation was performed using Research Center for Computational Science, Okazaki, Japan (Project: 22-OMS-C084, 21-IMS-C082). This study was supported by JSPS KAKENHI Grant Number JP17H06433 and JP17H06435 (to T.N.).

## Author contributions

T.N. designated the study. R.N. constructed the site-directed mutants and Y.S. isolated a histidine-tolerant strain. Y.S, T.M., and T.K. prepared samples and performed O$_2$ evolution and FTIR analyses. T.S. and N.D. performed LC-MS analysis. T.N. performed QM/MM calculations. T.N. wrote the first draft, and Y.S., T.S., T.M., T.K., R.N., N.D., and T.N. completed the manuscript.

## Competing interests

The authors declare no competing interests.
