## [Peer Review File · Nature Communications]

Post-translational amino acid conversion in photosystem II as a possible origin of photosynthetic oxygen evolutionEditorial Note: This manuscript has been previously reviewed at another journal that is not operating a transparent peer review scheme. This document only contains reviewer comments and rebuttal letters for versions considered at Nature Communications.

Reviewers' Comments:

Reviewer #1:

Remarks to the Author:

I had provided detailed feedback to the first version of this manuscript by Shimida et al., with emphasis on the evolutionary aspects of photosystem II. I am quite satisfied with the revised manuscript. A lot of my input was taken on board and therefore I do not have any additional criticism, just a few minor comments:

Line 58. I suggest the following amendment: "existed at a homodimeric photosystem stage..."

Line 81-86. When you say that "no trace of a codon for ASP was detected", it sounds as if this was a quantitative measurement. Is it the case? Please address.

Lines 128-135. Mn-depletion. There is no detail in your Methods section on how Mn depletion was carried out. I have found recently that even when not adding Mn into the BG11, that there's still enough contamination from other compounds or glassware to allow photosynthetic growth. Do you do any acid washing of your glassware, or have you used particularly high-grade compounds for the BG11? Please detail.

It also reads as if a single Mn atom is sufficient to allow the post-translational modification: see line 128 and 165. How can you rule out that a partial assembled cluster is not involved with two or three Mn atoms? Please rephrase or discuss.

Line 219. In this context, do you consider 100 umoles photons/m²/s low light? It is not clear when you consider what is written in the Methods. Please amend.

Section starting line 234. Quite nice and balanced evolutionary considerations without getting into tricky territory. Excellent.

Reviewer #2:

Remarks to the Author:

The work by Shimada, Y. et al. reports an interesting phenomenon of posttranslational conversion of amino acid residues around the Mn cluster in cyanobacterial PSII. A hypothesis is proposed by the authors, suggesting that the conversion may account for the formation of a primitive form of Mn cluster capable of partial water oxidation in ancestral PSII during the evolution. While the manuscript has been improved during the last revision, there are some minor points to be addressed by the authors.

(1) The authors have included a short paragraph describing the RNA sequencing result in lines 81-86, p4. It will be great if they could add an extended data figure showing the result and cite it in the text.

(2) In lines 172-187 (p7-p8), the authors discussed the mechanisms of metal-catalyzed conversion of His and Gln/Asn residues. Adding a cartoon model figure summarizing the mechanisms of the two

different chemical reactions will be helpful for the readers to understanding them better.

(3) In the last section (p10-p11), the discussion on the putative roles of posttranslational amino acid conversion in the evolution of photosynthetic water oxidation is mostly speculative. It might be better if the authors put it under a "Discussion" section.

(4) Are there any implications of the posttranslational amino acid conversion on the photodamage and photoinhibition of PSII besides the evolutionary insight?

(5) The work presents experimental evidences on the formation of native-like OEC in the site-directed D1 mutants (D170H, E189Q and D342N) through posttranslational conversion, whereas the point on the origin of photosynthetic oxygen evolution is a hypothesis instead of a conclusion. Therefore, the title "Origin of photosynthetic oxygen evolution beyond the central dogma" needs to be revised to reflect the major conclusion supported by the evidences, not the hypothesis.

(6) Line 35, "... leads to the formation ...". In the sentence, a past tense ("led") might be more appropriate.

Reviewer #3:

Remarks to the Author:

The authors have very carefully revised the manuscript taking into account the comments of all reviewers. This is an excellent and highly original work, and the proposal regarding evolution is solidly motivated by the reported data. All aspects of this work appear to have been performed with great care and competence. I strongly support publication in its present form.

Response to Reviewer #1

I had provided detailed feedback to the first version of this manuscript by Shimida et al., with emphasis on the evolutionary aspects of photosystem II. I am quite satisfied with the revised manuscript. A lot of my input was taken on board and therefore I do not have any additional criticism, just a few minor comments:

Author reply 1: We would like to thank the reviewer for his positive comment about the evolutionary aspects in the manuscript. We have revised the manuscript following your additional comments. We hope that the manuscript is now in the form publishable in Nature Communications.

Line 58. I suggest the following amendment: “existed at a homodimeric photosystem stage...”

Author reply 2: We incorporated “photosystem” in this sentence (p. 3, line 20).

Line 81-86. When you say that “no trace of a codon for ASP was detected”, it sounds as if this was a quantitative measurement. Is it the case? Please address.

Author reply 3: We examined the traces of the GAT/GAC codons for Asp in the cDNA of *psbA2* mRNA by checking the shape of the DNA sequence chromatogram. We confirmed that there was no band of G at the C position of the CAT codon for His170. In the revised manuscript, we showed the raw data of the DNA sequence chromatogram as Supplementary Fig. 1 and cited it in the text (p. 4, line 17).

Lines 128-135. Mn-depletion. There is no detail in your Methods section on how Mn depletion was carried out. I have found recently that even when not adding Mn into the BG11, that there’s still enough contamination from other compounds or glassware to allow photosynthetic growth. Do you do any acid washing of your glassware, or have you used particularly high-grade compounds for the BG11? Please detail.

Author reply 4: Indeed, the BG-11 medium prepared simply removing MnCl₂ contains high Mn²⁺ contamination and hence cyanobacterial cells grown in this medium show high O₂ evolution. We checked the Mn²⁺ content in metal solutions used in preparation of BG-11 by ICP-MS and found that the Mn²⁺ contamination comes from ammonium ferric citrate. We thus replaced ammonium ferric citrate with FeSO₄·7H₂O (special grade) and citric acid. Photoactivation of the Mn₄CaO₅ cluster was suppressed in cells grown in thus prepared Mn²⁺-depleted BG-11 medium. In the revised text, we added the procedure of the preparation of the Mn²⁺-depleted BG-11 medium in the Method section (p. 11, lines 22-26):

“For cell growth in a Mn²⁺-depleted medium, the BG-11 medium in the absence of MnCl₂·4H₂O was prepared with high-grade reagents. In addition to removing MnCl₂, we replaced ammonium ferric citrate, which was found to contain high Mn²⁺ contamination, in the original BG-11 medium with FeSO₄·7H₂O (special grade) and citric acid.”

It also reads as if a single Mn atom is sufficient to allow the post-translational

modification: see line 128 and 165. How can you rule out that a partial assembled cluster is not involved with two or three Mn atoms? Please rephrase or discuss.

Author reply 5: As the reviewer suggested, more than one Mn ion can be involved in the process of post-translational amino acid conversion. In the revised manuscript, the sentences that mention the involvement of a Mn ion were changed to suggest the possible involvement of more than one Mn ion (p. 6, lines 1 and 7; p. 7, line 10).

Line 219. In this context, do you consider 100 $\mu\text{mol photons/m}^2/\text{s}$ low light? It is not clear when you consider what is written in the Methods. Please amend.

Author reply 6: The authors of ref. 40 designated the light with the intensity of 100 $\mu\text{mol photons m}^{-2} \text{ s}^{-1}$ as “low light” and the light with 1000 $\mu\text{mol photons m}^{-2} \text{ s}^{-1}$ as “high light”. As the reviewer pointed out, this expression is controversial to our light condition of $\sim 50 \mu\text{mol photons m}^{-2} \text{ s}^{-1}$ as full illumination in comparison with dim light with $\sim 5 \mu\text{mol photons m}^{-2} \text{ s}^{-1}$. In the revised manuscript, the sentence that mentioned the result of ref. 40 was modified to remove the discrepancy:

(p. 9, line 4) “..... was previously shown to occur in several hours under the light of 100 $\mu\text{mol photons m}^{-2} \text{ s}^{-1}$ 40.”

(p. 9, line 7) “.... has time constants of hours under the light of $\sim 50 \mu\text{mol photons m}^{-2} \text{ s}^{-1}$ used in our experiments.”.

Section starting line 234. Quite nice and balanced evolutionary considerations without getting into tricky territory. Excellent.

Author reply 7: We truly appreciate the valuable comments of the reviewer #1 to our first manuscript, providing an insight into the evolutionary aspects of our finding of post-translational amino acid conversion in OEC.

Response to Reviewer #2

The work by Shimada, Y. et al. reports an interesting phenomenon of posttranslational conversion of amino acid residues around the Mn cluster in cyanobacterial PSII. A hypothesis is proposed by the authors, suggesting that the conversion may account for the formation of a primitive form of Mn cluster capable of partial water oxidation in ancestral PSII during the evolution. While the manuscript has been improved during the last revision, there are some minor points to be addressed by the authors.

Author reply: We would like to thank the reviewer for his valuable comments to improve the manuscript. We addressed all the points (see below), and we hope that the manuscript is now in the form publishable in Nature Communications.

(1) The authors have included a short paragraph describing the RNA sequencing result in lines 81-86, p4. It will be great if they could add an extended data figure showing the result and cite it in the text.

Author reply 1: We added a supplementary figure (Supplementary Fig. 1) showing the raw traces in the DNA sequence chromatogram of the cDNA of *psbA2* mRNA and cited it in the text (p. 4, line 17).

(2) In lines 172-187 (p7-p8), the authors discussed the mechanisms of metal-catalyzed conversion of His and Gln/Asn residues. Adding a cartoon model figure summarizing the mechanisms of the two different chemical reactions will be helpful for the readers to understanding them better.

Author reply 2: We added a figure (new Fig. 4) to summarize the possible mechanisms of amino acid conversions from His to Asp and from Asn/Gln to Asp/Glu, and cited this figure in the text (p. 7, line 26 and p. 8, line 2).

(3) In the last section (p10-p11), the discussion on the putative roles of posttranslational amino acid conversion in the evolution of photosynthetic water oxidation is mostly speculative. It might be better if the authors put it under a "Discussion" section.

Author reply 3: We have put the speculative discussion about the origin of photosynthetic oxygen evolution into the Discussion section (p. 9, line 19 – p. 10, line 20).

(4) Are there any implications of the posttranslational amino acid conversion on the photodamage and photoinhibition of PSII besides the evolutionary insight?

Author reply 4: Photodamage or photoinhibition of PSII takes place either by the attack of singlet oxygen that is produced from the triplet chlorophyll at Chl_{D1} when quinone electron acceptors are over reduced (acceptor side mechanism) or by oxidative damage to the electron donor side (mostly likely Y_Z and the surroundings of P680) by the high oxidation power of a P680 cation when the Mn cluster is inactivated (donor side mechanism). In either case, post-translational amino acid conversion at the ligand sites of the Mn cluster may not be related to the protection mechanism against the damage.

(5) The work presents experimental evidences on the formation of native-like OEC in the site-directed D1 mutants (D170H, E189Q and D342N) through posttranslational conversion, whereas the point on the origin of photosynthetic oxygen evolution is a hypothesis instead of a conclusion. Therefore, the title "Origin of photosynthetic oxygen evolution beyond the central dogma" needs to be revised to reflect the major conclusion supported by the evidences, not the hypothesis.

Author reply 5: The title was changed to “Post-translational amino acid conversion in photosystem II as a possible origin of photosynthetic oxygen evolution” to emphasize more the experimental finding of post-translational amino acid conversion in PSII.

(6) Line 35, "... leads to the formation ...". In the sentence, a past tense ("led") might be more appropriate.

Author reply 6: “leads” was changed to “led” in this sentence (p. 2, line 11).

Response to Reviewer #3

The authors have very carefully revised the manuscript taking into account the comments of all reviewers. This is an excellent and highly original work, and the proposal regarding evolution is solidly motivated by the reported data. All aspects of this work appear to have been performed with great care and competence. I strongly support publication in its present form.

Author reply: We sincerely appreciate the reviewer's positive comment. We hope that the manuscript revised in response to the other reviewers' comments is now in an acceptable form in Nature Communications.

Other format changes

1. Abstract was shortened to less than 150 words.
2. Results and Discussion sections were divided and the subtitles in the Discussion section were removed.
3. Supplementary Information was put in a separated PDF file.

Reviewers' Comments:

Reviewer #1:

Remarks to the Author:

All my comments have been addressed satisfactorily. I have no further comments or criticism.

Reviewer #2:

Remarks to the Author:

The authors have addressed my previous questions satisfyingly and improved their manuscript constructively. I have no further questions.

A point-by-point response to the reviewers' comments

Response to Reviewer #1

All my comments have been addressed satisfactorily. I have no further comments or criticism.

Author reply: We would like to thank the reviewer for his final comment, and we appreciate his previous valuable comments to improve the manuscript.

Response to Reviewer #2

The authors have addressed my previous questions satisfyingly and improved their manuscript constructively. I have no further questions.

Author reply: We appreciate the reviewer's previous comments to improve the manuscript.